# APT-36K: A Large-scale Benchmark for Animal Pose Estimation and Tracking

**Yuxiang Yang**[1], **Junjie Yang**[1], **Yufei Xu**[2†], **Jing Zhang**[2†], **Long Lan**[3], **Dacheng Tao**[4,2]

[1] School of Electronics and Information, Hangzhou Dianzi University, Hangzhou 310018, China
[2] School of Computer Science, The University of Sydney, NSW 2006, Australia
[3] National University of Defense Technology, Changsha 410073, China
[4] JD Explore Academy, Beijing 101111, China
yyx@hdu.edu.cn, 212040105@hdu.edu.cn, yuxu7116@uni.sydney.edu.au
jing.zhang1@sydney.edu.au, long.lan@nudt.edu.cn, dacheng.tao@gmail.com

## Abstract

Animal pose estimation and tracking (APT) is a fundamental task for detecting and tracking animal keypoints from a sequence of video frames. Previous animal-related datasets focus either on animal tracking or single-frame animal pose estimation, and never on both aspects. The lack of APT datasets hinders the development and evaluation of video-based animal pose estimation and tracking methods, limiting real-world applications, e.g., understanding animal behavior in wildlife conservation. To fill this gap, we make the first step and propose APT-36K, i.e., the first large-scale benchmark for animal pose estimation and tracking. Specifically, APT-36K consists of 2,400 video clips collected and filtered from 30 animal species with 15 frames for each video, resulting in 36,000 frames in total. After manual annotation and careful double-check, high-quality keypoint and tracking annotations are provided for all the animal instances. Based on APT-36K, we benchmark several representative models on the following three tracks: (1) supervised animal pose estimation on a single frame under intra- and inter-domain transfer learning settings, (2) inter-species domain generalization test for unseen animals, and (3) animal pose estimation with animal tracking. Based on the experimental results, we gain some empirical insights and show that APT-36K provides a valuable animal pose estimation and tracking benchmark, offering new challenges and opportunities for future research. The code and dataset will be made publicly available at https://github.com/pandorgan/APT-36K.

## 1 Introduction

Pose estimation aims to identify the categories and distinguish the locations of a series of body keypoints from an image. As a fundamental task in computer vision, it is beneficial for many vision tasks [28, 4, 3, 43] like behavior understanding, action recognition, etc. There has been rapid progress in human pose estimation thanks to the availability of a large number of human pose datasets [24, 21, 2]. However, fewer works focus on animal pose estimation, especially for video-based animal pose estimation, although it is crucial in animal behavior understanding and wildlife conservation.

Some efforts have been made to establish animal pose estimation datasets to facilitate research in this area. Early works focus on the pose estimation of specific animal categories, *e.g.*, the horse [26], zebra [12], macaque [19], fly [29], and tiger [22] datasets collect and annotate the keypoints for some

---

[†]Corresponding authors.

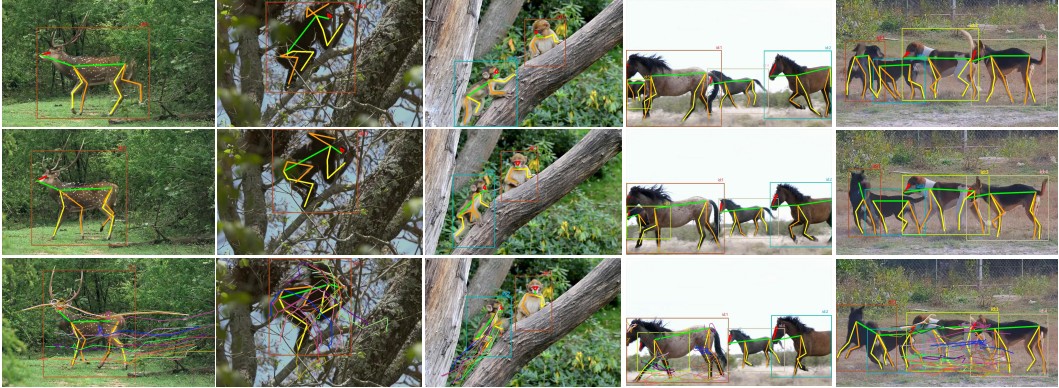

Figure 1: A glance of some examples from the proposed APT-36K dataset. The five columns of images show single animal instance, complex movements, and multiple animal instances, respectively. The keypoint trajectories of some animal instances are shown in the last frames. Best viewed in color.

specific kind of animals. They do help advance the research of pose estimation for these animals. However, since there exist huge appearance variance, behavior difference, and joint distribution shifts among various animal species due to evolution, the models trained on these datasets do not perform well on unseen animal species, leading to poor generalization performance. To further facilitate the research on animal pose estimation, datasets covering multiple animal species with keypoint annotations are proposed, *e.g.*, Animal-Pose [7] and AP-10K [40]. Despite their large scale in dataset volume and diversity in animal species, they lack important temporal information, making it unexplored to recognize poses in contiguous frames, which is very important for animal action recognition and beyond.

To fill this gap, we propose APT-36K, i.e., the first large-scale dataset with high-quality animal pose annotations from consecutive frames for animal pose estimation and tracking. APT-36K consists of 2,400 video clips collected and filtered from 30 different animal species with 15 frames for each video, resulting in 36,000 frames in total. These animals can be further classified into 15 animal families following taxonomic rank to facilitate the evaluation of inter-species and inter-family generalization ability of pose estimation models. Specifically, the video clips are collected from YouTube videos with carefully filtering. Then, the frames are sampled with specific intervals (i.e., at a low frame-per-second (FPS) speed) to remove duplication and increase the temporal motion amplitude. Next, 16 well-trained annotators are recruited to label the keypoints for each animal in each frame following the labeling routines of MS COCO [24], which are then manually double-checked. The trajectory of each animal is also denoted with bounding boxes and exclusive instance ids across the videos. In this way, APT-36K can support the research on both single-frame pose estimation and animal estimation tracking in consecutive frames.

Based on APT-36K, we set up three tracks to benchmark previous state-of-the-art (SOTA) pose estimation methods [32, 36, 41, 37, 38], *i.e.*, (1) single-frame animal pose estimation (SF track), (2) inter-species domain generalization (IS track), and (3) animal pose tracking (APT track). In the basic SF track, we comprehensively evaluate the performance of representative convolutional neural networks (CNN) and vison transformer-based methods in different settings, including inter- and intra-domain transfer learning, where the models are pre-trained on ImageNet dataset [10], MS COCO human pose estimation dataset [24], and AP-10k animal pose estimation dataset [40], respectively. In the IS track, the inter-family domain generalization ability of different pose estimation methods is evaluated, where the model is trained on images of all species from a certain family and tested on the images of other families. In the APT track, several object trackers, including a customized one based on the plain vision transformer [11], are used to track the animal instances, and representative pose estimation methods are used to detect the keypoints of the tracked instances, where their performance is evaluated accordingly. The detailed experiment settings and results are presented in Section 4, from which we demonstrate that the great potential of vision transformers in both animal pose estimation and animal pose tracking, the benefits of knowledge transfer between human pose estimation and

animal pose estimation, as well as the advantages of employing diverse animal species for animal pose estimation.

The main contribution of this paper is two-fold. First, we establish the first large-scale benchmark APT-36K for animal pose estimation and tracking. Its large scale, diversity of animal species, and abundant annotations of keypoints, bounding boxes, and instance ids among consecutive frames make it a good test bed for future study. Second, we set up three challenging tasks, including SF, IS, and APT, based on APT-36K and comprehensively benchmark representative pose estimation methods built upon both CNN and vision transformers, gaining some useful insights.

## 2 Related work

### 2.1 Human pose estimation

Pose estimation is a fundamental computer vision task for many applications such as behavior understanding. In the past decades, significant progress has been made in human pose estimation no matter in datasets [24, 21, 2] and methods [36, 32, 41, 37, 42]. For example, MPII [2] and MS COCO [24] are two popular large-scale benchmarks for human pose estimation. To further evaluate the performance of human pose estimation methods regarding challenging scenarios such as occlusion or crowd, OCHuman [46] and CrowdPose [21] are established where there are heavy occlusions of body keypoints or multiple human instances in a single frame. Both top-down and bottom-up methods have been proposed and evaluated based on these datasets. Despite the significant contribution made by these works, the temporal information has been largely ignored, which is important to understanding human behavior and action imitation from humans to robots. To address this issue, several video-based pose estimation datasets have been proposed, *e.g.*, VideoPose [31], YouTube Pose [8], J-HMDB [16], and PoseTrack [1]. They facilitate the research of human pose estimation and tracking [36].

### 2.2 Animal pose estimation

Recently, animal pose estimation has attracted increasing attention from the research community due to animal behavior understanding and wildlife conservation demand. Generally, animal pose estimation methods share similar ideas to human pose estimation ones, *e.g.*, bottom-up and top-down methods based on heatmap regression [40]. Nevertheless, different from the human pose, where the appearance, movement pattern, and keypoint distribution are similar among different people, they vary significantly for different animal species due to the difference in their habitat and evolutionary route. To facilitate the research in this area, many datasets have been proposed. In early works, single category animal pose estimation datasets are introduced, *e.g.*, horse [26], zebra [12], macaque [19], fly [29], and tiger [22] datasets. However, models trained on these datasets suffer from a limited generalization ability due to the significant difference in appearance and movement pattern between different animal species. To address this issue, some datasets covering many animal species have been established, including Animal-Pose [7], Animal Kingdom [27], and AP-10K [40]. For example, AP-10K contains 10,015 annotated images from 23 animal families and 54 species. Nevertheless, there are no temporal annotations in these datasets, making it impossible to develop animal pose tracking methods. Recently, a dataset named AnimalTrack for animal tracking has been established [44]. However, it only focuses on animal instance tracking rather than the fine-grained keypoint tracking. Besides, it has only limited video clips and animal species, *e.g.*, fewer than 60 video clips and 10 animal categories. Different from the above works, we propose APT-36K to fill the gap of the lack of real-world animal pose tracking datasets. Thanks to its large scale, diversity of animal species, and abundant annotations of keypoints, bounding boxes, and instance ids, we believe our APT-36K will benefit the research of animal pose estimation and tracking by serving as a training data source as well as a test bed along with several well-defined benchmark tracks.

### 2.3 Visual object tracking

Object tracking [20, 28, 15, 18] is a fundamental and active research topic in computer vision. One popular direction for object tracking follows the tracking by detection routine. For example, given the current frame and subsequent frames, an object detector is first used to detect the candidates from subsequent frames. Then, different techniques are employed to associate the detection results

with the target in the current frame. These methods obtain superior results in both multiple object tracking (MOT) [34, 47] and single object tracking (SOT) [5, 48, 33]. However, the generalization abilities of these trackers are limited, *i.e.*, the objects they can track should belong to the categories that the detectors support. The limitation hinders their usage in animal pose tracking, where many animal species may be unseen during the training of the detectors. The other development direction of object tracking follows the tracking by matching pipeline, *i.e.*, a siamese network is utilized to extract features from the tracked targets in the previous frame and the search regions in the subsequent frame, and then the two kinds of features are matched to localize the targets in the search region. In this paper, we mainly adopt this kind of tracking method for animal instance tracking since they do not make assumptions about the target categories. Besides, we benchmark their performance for animal pose tracking by combining them with different animal pose estimation methods.

## 3 Dataset

In this section, we briefly introduce the details of the proposed APT-36K dataset, including data collection and organization, data annotation, and the statistics of the dataset. Moreover, we provide detailed datasheets and more results in the supplementary material.

### 3.1 Data collection and organization

The goal of APT-36K is to provide a large-scale benchmark for animal pose estimation and tracking in real-world scenarios, which has been rarely explored in prior art. To this end, we resort to real-world video websites, *i.e.*, YouTube, and carefully collect and filter 2,400 video clips covering 30 different animal species from different scenes, *e.g.*, zoo, forest, and desert. However, directly annotating these videos and using them as training data is not appropriate since the movement speed of different animals and the frame frequency of different videos vary a lot, *e.g.*, some animals are almost static during a specific period. To address this issue, we manually set the frame sampling rate for each video to ensure there are noticeable movement and posture differences for each animal in the sub-sampled video clips. Specifically, each clip contains 15 frames after the sampling process. It should be noted that challenging cases such as truncation and occlusion are kept in the dataset owing to the above process, making it possible to evaluate the models regarding these challenges.

After the video collection and cleaning process, we categorize the videos from 30 animal species further into 15 families following the taxonomic rank. Following the terms of YouTube, we use sparsely sampled frames in the videos to formulate the APT-36K dataset and use them for research purposes only. According to Linnean's theory of evolution, animals belonging to the same taxonomic rank may share more similarities in behavior patterns, anatomical keypoint distribution, and appearance than those belonging to different families. For example, the walking posture of dogs and wolves is similar since they belong to the Canidae family, while zebras' walking patterns are far from similar to them since it belongs to a different family, *i.e.*, Equidae. Following the taxonomic rank, the proposed dataset can be easily scaled up by collecting and annotating more animal images from the same species or families, as well as different ones. Moreover, it also implies that such an organized way of the animal pose dataset provides a possible way to enhance the generalization ability of animal pose estimation models to rare animal species, *i.e.*, by collecting and annotating images from other more common animals of the same taxonomic rank.

### 3.2 Data annotation

To guarantee high-quality annotations for each image in the APT-36K dataset, 16 well-trained annotators participated in the annotation process, and one strict cross-check was then carried out to improve the annotation quality. The annotation-check round is repeated three times during the labeling process. The whole data collection, cleaning, annotation, and check process takes about 2,000 person-hours. A total of 36,000 images are finally labeled, following the COCO labeling format. There are typically 17 keypoints labeled for each animal instance, including two eyes, one nose, one neck, one tail, two shoulders, two elbows, two knees, two hips, and four paws as [40]. It should be noted that we do not exactly follow the biological definition to localize the keypoints for specific animals, *i.e.*, we use the paw to define the tire point of horses and the knee to represent the end of their hock. In this way, it helps us better figure out the behavior of specific animals since half of the horses' legs will have no annotations if we strictly follow the anatomy definition. Besides

the keypoint annotations, we label the background type for each frame from 10 classes, *i.e.*, grass, city, and forest. In addition, we label each same animal instance across the video clips with a unique tracking id. The annotations are also manually checked for two rounds to improve their quality. The dataset is split into three disjoint subsets for training, validation, and test, respectively, following the ratio of 7:1:2 per animal species. It is also noteworthy that we adopt a video-level partition to prevent the potential information leakage since the frames in the same video clip are similar to each other.

## 3.3   Statistics of the APT-36K dataset

Table 1: Comparison of different animal pose datasets.

|  | #Species | #Family | #Labeled image | #Keypoint | #Sequence | #Instance | #Background type |
|---|---|---|---|---|---|---|---|
| Horses-10 [26] | 1 | 1 | 8,100 | 22 | N/A | 8,110 | N/A |
| Animal-Pose Dataset [7] | 5 | N/A | 4,666 | 20 | N/A | 6,117 | N/A |
| Animal kingdom [27] | 850 | 6 | 33,099 | 23 | N/A | N/A | 9 |
| AP-10K [40] | 54 | 23 | 10,015 | 17 | N/A | 13,028 | N/A |
| Animal track [44] | 10 | N/A | 24,700 | N/A | 58 | 429,000 | N/A |
| APT-36K | 30 | 15 | 36,000 | 17 | 2,400 | 53,006 | 10 |

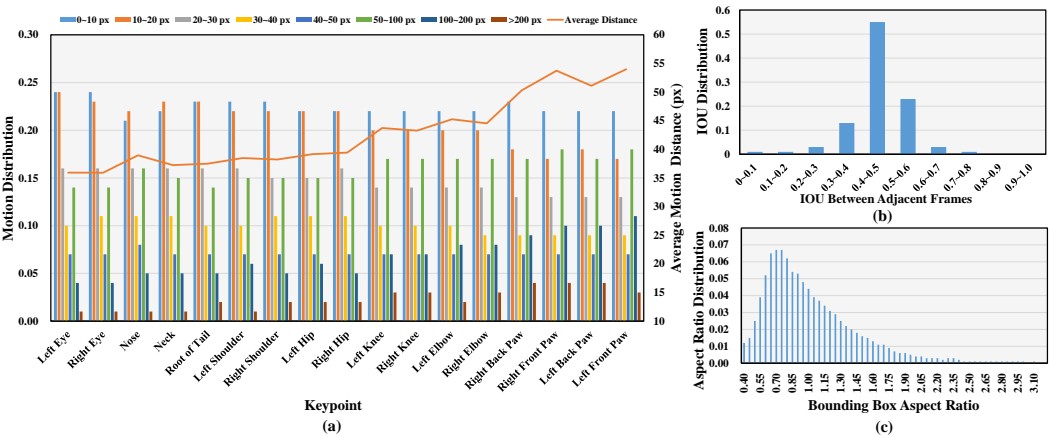

Figure 2: Statistics of APT-36K. (a) The motion distribution and average motion distance of different keypoint categories. "N/A" indicates that the corresponding entry is not available in the given dataset. (b) The distribution of IOU scores between the tracked bounding boxes in adjacent frames. (c) The distribution of aspect ratios of the bounding boxes in APT-36K.

As shown in Table 1, APT-36K contains 30 different animal species belonging to 15 different families. It has 36,000 annotated frames with 53,006 annotated animal instances from 2,400 video clips, which are much richer than previous animal pose estimation datasets. Therefore, it sets new challenges for animal pose estimation tasks. All the videos are collected from the YouTube website from a set of different topics, including documentary films, vlogs, education films, *etc*.They are captured using different cameras, at different shooting distances with diverse camera movement patterns. There are 10 types of background in the images of APT-36K, providing diverse scenes for a comprehensive evaluation of animal pose estimation and tracking. For each animal species in APT-36K, there are 80 video clips in total, making it a balanced dataset. It is different from previous animal pose estimation datasets like AP-10K, which is long-tailed and has much fewer instances in some species, *e.g.*, Cercopithecidae. APT-36K is the first dataset suitable for both animal pose estimation and tracking, filling the gap in this area and providing new challenges and opportunities for future research.

We also calculate the distributions of the keypoint motion, IOU between tracked bounding boxes in adjacent frames, and the aspect ratio of the annotated bounding boxes in our APT-36K dataset. As shown in Figure 2 (a), the motion distribution and average motion distance vary a lot for different keypoints, *e.g.*, the average motion distance of paws is over 50 pixels, which is much larger than that of eyes or necks (about 35 pixels). Moreover, the motion magnitudes of shoulder, knee, and hips lie between those of eyes and paws, which is in line with the movement characteristics of four-leg animals. Besides, most of the instances have small IOU scores between their tracked bounding boxes

in adjacent frames, implying large motion is very common in APT-36K, as demonstrated in Figure 2 (b). It can also be observed from Figure 2 (c) that the aspect ratio of the bounding box varies a lot from less than 0.4 to more than 3.1. It is because APT-36K contains diverse animals with different actions, *e.g.*, running rabbits and climbing monkeys. These results illustrate the diversity of APT-36K.

## 4 Experiment

### 4.1 Implementation details

To provide a comprehensive evaluation for animal pose estimation and tracking, we benchmark representative CNN-based and vision transformer-based pose estimation methods [32, 36, 37, 41] using ground truth bounding boxes or tracked object boxes on the proposed APT-36K dataset. Representative tracking methods [39, 20, 23] are employed to obtain the tracked boxes for the animal instances in the video clips. We set up three tracks based on APT-36K, *i.e.*, the SF track, IS track, and APT track, as described in Sec. 1. The models are implemented based on the MMPose [9] codebase and trained for 210 epochs following the common practice in human/animal pose estimation tasks. The initial learning rate is 5e-4 and decreased by a factor of 10 at the 170th and 200th epochs. The detailed experimental settings for each track are presented in the following part. We use the average precision (AP) [24] as the primary metric to evaluate the performance of different models. It can be calculated as $AP = \frac{\sum_p \theta(oks_p > T)}{\sum_p 1}$, where $p$ is the index of the person and $oks_p$ is the object keypoint similarity metric. The OKS metric is defined as $\sum_i [exp(-d_i^2/2s^2k_i^2)\theta(v_i > 0)]/\sum_i [\theta(v_i > 0)]$, where $d_i$ is the distance between the $i$-th predicted results and the $i$-th ground truth keypoint locations. $s$ is the scale of the object and $k_i$ is a predefined constant that controls falloff. $v_i$ indicates the visibility of the $i$-th keypoint. We use the constant as defined in AP-10K [40].

### 4.2 Single-frame animal pose estimation (SF track)

**Setting** In the SF track, we benchmark the representative CNN-based and vision transformer-based pose estimation methods, including SimpleBaseline [36], HRNet [32], HRFormer [41], and ViTPose [37]. The SimpleBaseline takes the ResNet [14] (*i.e.*, ResNet-50 and ResNet-101) as the backbone encoder for feature extraction and uses three deconvolution blocks to up-sample the feature maps for decoding. HRNet takes a similar pipeline but employs a multi-resolution parallel backbone network to extract high-resolution feature maps and discards the deconvolution blocks in the decoder part. HRFormer follows a similar spirit to HRNet and takes a multi-stage transformer structure with multiple branches to jointly encode the multi-resolution information into a high-resolution feature map. ViTPose, on the other hand, utilizes a plain and non-hierarchical vision transformer as a backbone encoder for feature extraction. We set up three settings to benchmark their performance, including using network weights pre-trained on the ImageNet-1K dataset [10], the MS COCO human pose estimation dataset [24], and the AP-10K dataset [40], respectively. We randomly split the dataset three times with random seeds 0, 10,000, and 20,000, respectively, and train each model accordingly to estimate the error bar of their performance. Specifically, the three settings are detailed as below:

- *ImageNet-1K pre-training.* It is a common practice to use ImageNet-1K pre-trained weights to initialize the backbones of pose estimation models. We follow this practice and fine-tune the models initialized with ImageNet-1K pre-trained weights for further 210 epochs on the APT-36K training set. It is noted that we adopt the fully supervised learning scheme on ImageNet-1K to get the pre-trained weights for the backbones used by SimpleBaseline, HRNet, and HRFormer. The vision transformer backbones in ViTPose are initialized with the pre-trained weights from the self-supervised MAE pre-training [13]. The Adam [17] optimizer is utilized for training the CNN-based models, and AdamW [30] optimizer is employed to train the vision transformer-based ones, following their default settings.

- *Human pose pre-training.* Since the keypoint definition of four-foot animals is similar to those of human beings, it may be beneficial to transfer the knowledge learned from human keypoint annotations to animal pose estimation. Consequently, it will help us make use of the existing large-scale datasets for human pose estimation and bypass the difficulties of building large animal pose estimation datasets. To this end, we pre-train the CNN-based and vision transformer-based models on the MS COCO human pose estimation dataset for 210 epochs. Then, the pre-trained weights are used to initialize the models, which are

further fine-tuned on the APT-36K training set for another 210 epochs, following the above ImageNet-1K pre-training setting.

- *Animal pose pre-training.* Previous animal pose estimation datasets provide abundant animal images with keypoint annotations for four-foot animals. To investigate the benefit of leveraging the animal pose estimation datasets, we first pre-train the models on AP-10K for 210 epochs and further fine-tune them on the APT-36K training set for another 210 epochs.

Table 2: Results on the APT-36K val set (AP) of different models on the SF track with ImageNet-1K (IN1K) [10], MS COCO [24], and AP-10K [40] pre-training, respectively. $\Delta_{\text{COCO}}$ and $\Delta_{\text{AP}-10\text{k}}$ denote the gains of MS COCO and AP-10K pre-training over ImageNet-1K pre-training, respectively.

| | SimpleBaseline (ResNet-50) | SimpleBaseline (ResNet-101) | HRNet (HRNet-w32) | HRNet (HRNet-w48) | HRFormer (HRFormer-S) | HRFormer (HRFormer-B) | ViTPose (ViT-B) |
|---|---|---|---|---|---|---|---|
| IN1K | $69.4_{+1.2}$ | $69.6_{+1.3}$ | $74.2_{+1.1}$ | $74.1_{+0.8}$ | $71.3_{+0.8}$ | $74.2_{+0.9}$ | $77.4_{+1.0}$ |
| COCO | $73.7_{+1.2}$ | $73.5_{+1.1}$ | $76.4_{+0.5}$ | $77.4_{+0.7}$ | $74.6_{+1.0}$ | $76.6_{+0.9}$ | $78.3_{+0.8}$ |
| $\Delta_{\text{COCO}}$ | 4.3 | 3.9 | 2.2 | 3.3 | 3.3 | 2.4 | 0.9 |
| AP-10K | $72.4_{+0.9}$ | $72.4_{+1.0}$ | $75.9_{+1.2}$ | $76.4_{+0.7}$ | $72.6_{+0.9}$ | $75.2_{+0.7}$ | $78.2_{+0.7}$ |
| $\Delta_{\text{AP}-10\text{K}}$ | 3.0 | 2.8 | 1.7 | 2.3 | 1.3 | 1.0 | 0.8 |

**Results and analysis** The results are summarized in Table 2. It can be observed that with human pose pre-training, both CNN-based and vision transformer-based methods show performance gains, *e.g.*, from 69.6 AP to 73.5 AP for SimpleBaseline with a ResNet-101 backbone network, from 74.1 AP to 77.4 AP for HRNet-w48, and from 74.2 AP to 76.6 AP for HRFormer-B. A similar benefit can also be obtained by using AP-10K for pre-training, *e.g.*, SimpleBaseline reaches 72.4 AP with either a ResNet-101 or ResNet-50 backbone network, and HRNet-w48 gets a gain of 2.3 AP compared with the ImageNet-1K pre-training. Generally, the benefit is slightly more significant for models with worse performance than those stronger models, which is reasonable. It is noteworthy that ViTPose with a plain vision transformer backbone, which is pre-trained on ImageNet-1K without using its labels, obtains a remarkable performance of 77.4 AP. Also, after human pose pre-training or animal pose pre-training, the performance could be further improved to 78.3 AP and 78.2 AP, respectively, which is much better than other models.

Another interesting finding is that, although using the animal pose dataset AP-10K for pre-training brings performance gains, the benefit is less than that of using the human pose dataset for pre-training, no matter for CNN-based models or vision transformer-based models (See the third and fifth rows in Table 2 denoted by $\Delta_{\text{COCO}}$ and $\Delta_{\text{AP}-10\text{K}}$, respectively). We suspect there are two reasons. First, there is still a domain gap between AP-10K and APT-36K due to their different data sources and distributions, *i.e.*, imbalanced and long-tailed species in AP-10K v.s. balanced ones in APT-36K. Second, MS COCO is much larger than AP-10K by about an order of magnitude, probably leading to more sufficient pre-training since the model could see more diverse training instances and learn more discriminative feature representations. Nevertheless, the difference between $\Delta_{\text{COCO}}$ and $\Delta_{\text{AP}-10\text{k}}$ is not evident for ViTPose, owing to the strong representation ability of vision transformers.

### 4.3 Inter-species animal pose generalization (IS track)

**Setting** Regarding the diverse animal species in the real world, it is essential to evaluate the inter-species generalization ability of animal pose estimation models, *i.e.*, investigating their performance on unseen animal species. To this end, we set up the IS track, where we select six representative animal families for training and test, *i.e.*, Canidae, Felidae, Hominidae, Cercopithecidae, Ursidae, and Bovidae. In each experiment, all the instances from a specific family form the test set while those instances belong to other families are split into the training set and validation set at a ratio of 9:1. We use the representative HRNet-w32 model with MS COCO pre-training in this track due to its good performance and simplicity. The models are trained following the same setting in the SF track.

**Results and analysis** As can be seen from Table 3, the models generalize well on the Canidae, Felidae, Bovidae, and Equidae families with instances from other animals for training. For example, it obtains 57.6 AP on the Canidae family, as indicated in the top-left cell. It is because although the animal instances from the Canidae family are not used during training, they share some commonness with animals in the Ursidae family and Felidae family since they belong to the same Carnivora order. For rare species that do not share commonness with the training set, the models show much poorer generalization ability, *e.g.*, the model trained without data from the Cercopithecidae family only obtains 29.6 AP on the Cercopithecidae family. In contrast, after using the instances from the

Table 3: Results of HRNet-w32 models on the IS track (AP) of APT-36K.

| training \ test | Canidae | Felidae | Hominidae | Cercopithecidae | Ursidae | Bovidae | Equidae |
|---|---|---|---|---|---|---|---|
| *w/o* Canidae | _57.6_ | 82.8 | 81.9 | 76.0 | 80.1 | 81.6 | 84.0 |
| *w/o* Felidae | 80.9 | _57.6_ | 81.0 | 76.6 | 79.7 | 81.6 | 84.6 |
| *w/o* Hominidae | 80.3 | 81.9 | _44.2_ | 77.2 | 80.3 | 81.9 | 84.3 |
| *w/o* Cercopithecidae | 81.1 | 83.4 | 80.8 | _29.6_ | 80.7 | 81.1 | 84.7 |
| *w/o* Ursidae | 80.7 | 83.3 | 80.8 | 76.7 | _43.3_ | 82.0 | 84.1 |
| *w/o* Bovidae | 80.1 | 83.6 | 80.6 | 76.7 | 80.8 | _58.6_ | 84.5 |
| *w/o* Equidae | 80.4 | 82.4 | 82.0 | 77.2 | 79.7 | 82.2 | _61.5_ |

Cercopithecidae family for training, the performance could reach over 76.0 AP (See other scores in the fourth column except the diagonal one). It also should be noted that although the model generalizes well on the families mentioned above, the performance still falls behind the models that have seen data from those families during training by a large margin, *e.g.*, 57.6 AP v.s. over 80 AP on the Canidae family. The results imply that 1) each animal species has its own characteristics, 2) it is beneficial and necessary (if possible) to collect and annotate animal instances from diverse animal families, especially for rare species like the Cercopithecidae family, and 3) more efforts should be made to improve the inter-species generalization ability of animal pose estimation models. Besides, the models trained with slightly different training sets demonstrate similar performance on the same seen family as shown in each column (except the diagonal one) in Table 3, which is probably attributed to the balanced data distribution (80 video clips for each species) of animal species in the APT-36K dataset.

Table 4: Results of HRNet-w32 models on the few-shot learning setting (AP) of APT-36K.

| | Canidae | Felidae | Hominidae | Cercopithecidae | Ursidae | Bovidae | Equidae |
|---|---|---|---|---|---|---|---|
| zero-shot | 52.3 | 54.8 | 46.3 | 35.6 | 38.0 | 54.9 | 61.4 |
| 20-shot | 53.3 | 55.5 | 48.4 | 37.6 | 43.3 | 55.5 | 61.7 |
| 30-shot | 53.9 | 56.0 | 52.5 | 41.6 | 48.0 | 56.3 | 62.0 |
| 40-shot | 54.3 | 57.2 | 52.7 | 42.1 | 48.1 | 56.8 | 62.8 |

**Few-shot learning** To further evaluate the model's generalization ability, we carried out the experiment under the few-shot setting. As can be seen from Table 4, with more data used for training, the performance is greatly improved, especially on species with unique textures, appearance, and posture characteristics, e.g., the Hominidae, Cercopithecidae, and Ursidae species. It is because these unique characteristics are not shared in the training images of other species. The performance gain brought by more training data is relatively smaller for species that shares some common characteristics with the training species, e.g., the Canidae, Felidae, Bovidae, and Equidae species. We think that the few-shot setting is an important research topic in animal pose estimation, e.g., how to make the pre-trained models generalize well on unseen species. Besides, recent studies have shown that large models are already few-shot learners [45, 6]. It is interesting to explore their performance on the few-shot animal pose estimation tasks, where the proposed dataset can provide a suitable benchmark.

## 4.4 Animal pose tracking (APT track)

**Setting** In this track, we use representative object trackers with both CNN-based backbones and vision transformer-based backbones to track each animal instance across the video clips, giving each animal's ground truth bounding box in the first frame. Once the tracked bounding boxes are obtained, the pose estimation methods with MS COCO pre-training are used for animal pose estimation accordingly. We also use the average precision metric for evaluation. Specially, the CNN-based tracking methods SiamRPN++ [20] and STARK [39] with a ResNet-50 backbone [14] are employed for animal tracking. For vision transformer-based methods, we adopt the recent SOTA tracking method SwinTrack [23] with a Swin transformer backbone [25]. We also design a simpler **ViTTrack** baseline with the plain vision transformer as the backbone, *i.e.*, ViT [11], to compare the performance of both stage-wise transformer structure and plain transformer structure. ViTTrack is a siamese structure with a shared backbone encoder for feature extraction. The target object in the first frame is used as a template for feature matching in the subsequent frames and tracking. Specifically, the features from subsequent frames are concatenated with the template feature and fed into decoder layers, whose output is then used to predict the locations of the target object via a simple MLP. The encoder and decoder are all plain vision transformers, which are pre-trained on ImageNet-1K via

MAE [13]. To reduce the computational cost, the template image size is usually set to $112\times112$, while the size of the search region in the subsequent frames is set to $224\times224$.

Table 5: Results on the APT-36K test set (AP) of different models on the APT track with different object trackers. † denotes ViTTrack uses the fixed ViT encoder of ViTPose trained on APT-36K.

| | SimpleBaseline (ResNet-50) | SimpleBaseline (ResNet-101) | HRNet (HRNet-w32) | HRNet (HRNet-w48) | HRFormer (HRFormer-S) | HRFormer (HRFormer-B) | ViTPose (ViT-B) |
|---|---|---|---|---|---|---|---|
| SiamRPN++ [20] | $70.2_{\pm1.7}$ | $70.1_{\pm1.6}$ | $73.0_{\pm1.4}$ | $73.6_{\pm1.6}$ | $70.9_{\pm1.6}$ | $73.1_{\pm1.4}$ | $74.2_{\pm1.1}$ |
| STARK [39] | $71.5_{\pm1.7}$ | $71.4_{\pm1.7}$ | $74.1_{\pm1.4}$ | $74.8_{\pm1.5}$ | $72.1_{\pm1.5}$ | $74.2_{\pm1.4}$ | $75.3_{\pm1.0}$ |
| SwinTrack [23] | $71.6_{\pm1.8}$ | $71.5_{\pm1.6}$ | $74.1_{\pm1.4}$ | $74.9_{\pm1.6}$ | $72.2_{\pm1.8}$ | $74.3_{\pm1.5}$ | $75.4_{\pm1.1}$ |
| ViTTrack | $71.9_{\pm1.6}$ | $71.9_{\pm1.4}$ | $74.4_{\pm1.2}$ | $75.3_{\pm1.4}$ | $72.7_{\pm1.4}$ | $74.6_{\pm1.2}$ | $75.8_{\pm0.9}$ |
| ViTTrack† | $71.7_{\pm1.7}$ | $71.6_{\pm1.4}$ | $74.2_{\pm1.1}$ | $74.9_{\pm1.4}$ | $72.3_{\pm1.4}$ | $74.5_{\pm1.2}$ | $75.5_{\pm0.9}$ |

**Results and analysis** The results are summarized in Table 5. It can be observed that the APT performance based on the vision transformer-based trackers is slightly better compared with that using CNN-based trackers, *i.e.*, HRFormer-S [41] obtains 72.1 AP with STARK [39] and 72.7 AP with ViTTrack, respectively. Similarly, SimpleBaseline [36] with ResNet-50 [14] achieves 71.5 AP with STARK and 71.9 AP with ViTTrack. Again, ViTPose achieves the best performance among all the pose estimation models no matter which object tracker is used. Moreover, ViTPose with ViTTrack delivers the best 75.8 AP, surpassing the SimpleBaseline with SiamRPN++ tracker by a large margin of 5.6 AP. Besides, we initialize the ViT-B encoder of ViTTrack with the weights of the ViT-B encoder in ViTPose trained on APT-36K training set and keep it fixed during training on the tracking data, *i.e.*, denoted as ViTTrack† in Table 5. Surprisingly, with the shared ViT-B encoder, ViTPose with ViTTrack† obtains the second-best performance, *i.e.*, 75.5 AP, which is even better than ViTPose with SwinTrack, which requires an extra Swin transformer encoder. These results imply the potential of plain vision transformers as a foundation model for simultaneously serving multiple vision tasks, which is of great significance and deserves more research in future work.

Table 6: Results of different single object tracking methods (success and precision) on APT-36K.

| | SwinTrack [23] | SiamRPN [20] | STRAK [39] | ViTTrack | ViTTrack† |
|---|---|---|---|---|---|
| *Success* | 81.5 | 76.3 | 81.9 | 81.2 | 80.9 |
| *Precision* | 62.9 | 46.8 | 62.6 | 64.2 | 61.5 |

Multi-object tracking methods have been studied in previous pose-tracking tasks. However, it is really a hard job to adapt existing multi-object tracking methods to the animal pose tracking task, due to the scope mismatching issue between the animal species and in-distribution categories of current detectors. Specifically, most of the animals in the proposed APT-36K dataset can't be successfully recognized by the current object detectors. To address this issue, a simplified strategy in SimpleBaseline [36] using optical flow-based tracking is adopted to evaluate the performance of multi-object tracking on APT-36K. With the HRNet-w32 [32] pose estimator, this strategy obtains 71.3 AP, which is not well as the single-object tracking results, *e.g.*, at least 73.0 AP as shown in the 3rd column in Table 5. Thus, we focus on single-object tracking in this paper and compare their success rate and precision using the OPE metric [35]. The results are summarized in Table 6. It can be observed that the success rate of single object tracking on the APT-36K dataset is around 80 and the precision is slightly above 60 even for the strong tracker SwinTrack [23]. Such results demonstrate that the proposed APT-36K dataset is challenging for both multi-object and single-object tracking methods. How to efficiently deal with the diverse animal species deserves more research efforts in the future, *e.g.*, an object detector with better detection results for out-of-distribution categories.

## 5 Limitation and discussion

Our APT-36K fills the gap between single-frame animal pose estimation and animal tracking datasets, based on which we benchmarked representative pose estimation models and gained some insights. We believe APT-36K can benefit the future research of animal pose estimation and tracking, *e.g.*, regarding novel model design, effective inter-species generalization, and multi-task learning. Although the scale of APT-36K is larger than previous animal pose estimation datasets, it is still much smaller than human pose estimation datasets. Considering the diverse species of animals, there are more efforts to be made in future work. Besides, the number of average animal instances in the video clips of APT-36K is limited compared with that in animal instance tracking datasets. Although more animals in the video clips always imply many small and occluded instances, it makes the annotation process

of keypoint tracking extremely more difficult and expensive compared with annotating the bounding boxes of animal instances. To this end, it still matters to train better animal pose tracking models to be applicable in real-world scenarios. One possible solution is to resort to a modern 3D engine to generate realistic synthetic images and automatically produce annotations automatically.

# 6   Conclusion

We establish APT-36K, *i.e.*, the first large-scale dataset with high-quality animal pose annotations from consecutive frames for animal pose estimation and tracking. Based on APT-36K, we benchmark representative state-of-the-art pose estimation methods under the single-frame animal pose estimation setting, inter-species animal pose generalization setting, and animal pose tracking setting, respectively. Extensive experimental results demonstrate the benefit of inter- and intra-domain pre-training for animal pose estimation, the significance of collecting and annotating keypoints of diverse animal species, and the great potential of plain vision transformers for animal pose tracking. We hope APT-36K can provide new opportunities for further animal pose estimation and tracking research.

**Social impact.** The proposed APT-36K can facilitate the research of animal pose estimation and tracking, which is beneficial for animal behavior understanding and wildlife preservation. However, due to the great diversity of real-world animal species, it should be careful about whether a model trained on APT-36K can generalize well on unseen rare animal species.

## Acknowledgments and Disclosure of Funding

The project is funded by the National Natural Science Foundation of China under Grant 61873077.

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
