# APT-36K: A Large-scale Benchmark for Animal Pose Estimation and Tracking Supplementary Material

**Yuxiang Yang**[1], **Junjie Yang**[1], **Yufei Xu**[2], **Jing Zhang**[2], **Long Lan**[3], **Dacheng Tao**[4,2]

[1] School of Electronics and Information, Hangzhou Dianzi University, Hangzhou 310018, China
[2] School of Computer Science, The University of Sydney, NSW 2006, Australia
[3] National University of Defense Technology, Changsha 410073, China
[4] JD Explore Academy, Beijing 101111, China
yyx@hdu.edu.cn, 212040105@hdu.edu.cn, yuxu7116@uni.sydney.edu.au
jing.zhang1@sydney.edu.au, long.lan@nudt.edu.cn, dacheng.tao@gmail.com

## A  Subjective results

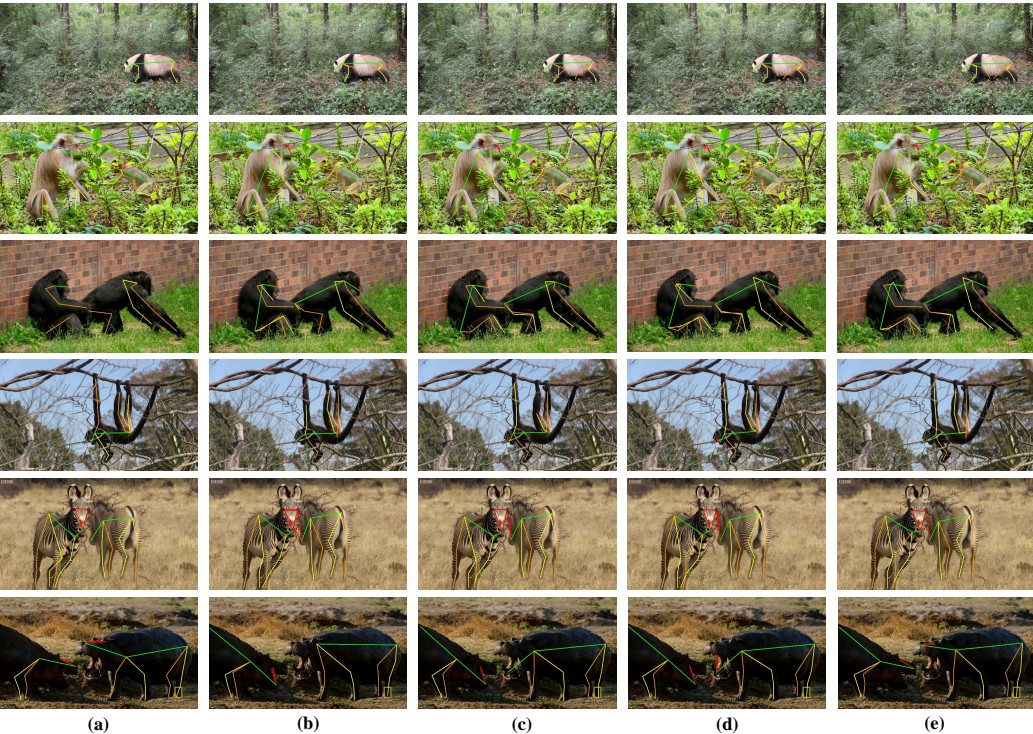

Figure 1: Subjective results of (a) groundtruth; (b) SimpleBaseline [7] with ResNet-101 [2]; (c) HRNet-w48 [6]; (d) HRFormer-B [10]; and (e) ViTPose [8]. The video-based results from ViTPose are provided in the supplementary video.

To subjectively evaluate the results of different pose estimation methods trained on the APT-36K dataset, we visualize the results generated by the representative methods SimpleBaseline [7] with

36th Conference on Neural Information Processing Systems (NeurIPS 2022) Track on Datasets and Benchmarks.

ResNet-101 [2] backbone, HRNet-w48 [6], HRFormer-B [10], and ViTPose [8] with ViT-B [1] backbone. All models are pre-trained with the human pose estimation data from the MS COCO [5] dataset. As demonstrated in Figure 1, the methods trained with the APT-36K dataset successfully predict the keypoints of different animal species, despite challenging cases like occlusion (the 2nd and 3rd row). They can also deal with animals with irregular body postures like in the 4th row and different orientations like in the 5th row. It can also be observed that ViTPose can better deal with challenging cases like multiple instances with irregular body postures. As shown in the last row of Figure 1, ViTPose predicts more precise eye and nose locations for the hippopotamus at the right.

Table 1: Detail animal species and classification information in APT-36K.

| Felidae | Bovidae | Canidae | Hominidae | Cercopithecidae | Ursidae | Equidae | Cervidae |
|---|---|---|---|---|---|---|---|
| cat | antelope | dog | chimpanzee | monkey | panda | horse | deer |
| tiger | buffalo | fox | orangutan | spider-monkey | black-bear | zebra | |
| cheetah | cow | wolf | gorilla | howling-monkey | polar-bear | | |
| lion | sheep | | | | | | |

| Leporidae | Suidae | Elephantidae | Hippopotamidae | Procyonidae | Rhinocerotidae | Giraffidae |
|---|---|---|---|---|---|---|
| rabbit | pig | elephant | hippo | raccoon | rhino | giraffe |

## B  Video clip choosing

To guarantee the diversity and annotation quality of the proposed APT-36K dataset, we select 30 different animal species with distinct features that people are familiar with to label. Animals that do not satisfy these requirements, such as hamsters and beavers, are not considered since these animals are indistinguishable when moving. To ensure a reasonable difficulty distribution of the dataset, we follows the 5:3:2 principles to select the videos, *i.e.*, 50% of the selected videos are the simple cases which contain a single animal without occlusion; 30% of them can be categorized as the medium-difficult cases which involves a single animal with occlusion; and the left 20% of them are difficult cases with multiple animals and occlusion. Meanwhile, to ensure the diversity of the backgrounds, we strictly control that there are fewer than two clips having the same kind of background.

## C  Annotators training

To train the annotators, we first employ animal skeletal models with annotation examples to teach the annotators how to annotate the keypoints for each animal. After that, we assign each annotator several test video clips for annotation. The annotation results on these test clips will be compared with the ground truth keypoint annotations to select the appropriate annotators in the following annotation process. During the annotation, we first have the annotators annotate each clip. The annotation results are checked by the senior annotators. After that, the checked results are further validated by the organizer to further reduce the errors in the annotations. To deal with challenging cases like occlusion, it should be noted that there are adjacent frames for a given frame to be annotated in a video. In this case, annotators can estimate the occluded keypoints via the clues from several adjacent frames, where the keypoints are visible and not-occluded. In addition, for the keypoints that are heavily occluded and can not be determined by the annotators with high confidence, we mark them as obscured and do not take them into account for model training and evaluation.

## D  Memory and inference speed comparison

Table 2: Training memory and inference speed comparison of SimpleBaseline [7], HRNet [6], HRFormer [10], and ViTPose [8] on the APT-36K dataset.

| | ResNet-50 [7] | ResNet-101 [7] | HRNet-w32 [6] | HRNet-w48 [6] | HRFormer-S [10] | HRFormer-B [10] | ViTPose-B [8] |
|---|---|---|---|---|---|---|---|
| Training Memory (M) | 10,041 | 13,589 | 15,051 | 19,635 | 28,888 | OOM | 19,715 |
| Throughput (FPS) | 1,102 | 834 | 719 | 521 | 208 | 123 | 711 |
| Precision (AP) | 73.7 | 73.5 | 76.4 | 77.4 | 74.6 | 76.6 | 78.3 |

We also compare these pose estimation methods' training memory footprint and inference speed on A100 machines with 40G memory. We use a batch size of 64 input images with a resolution of

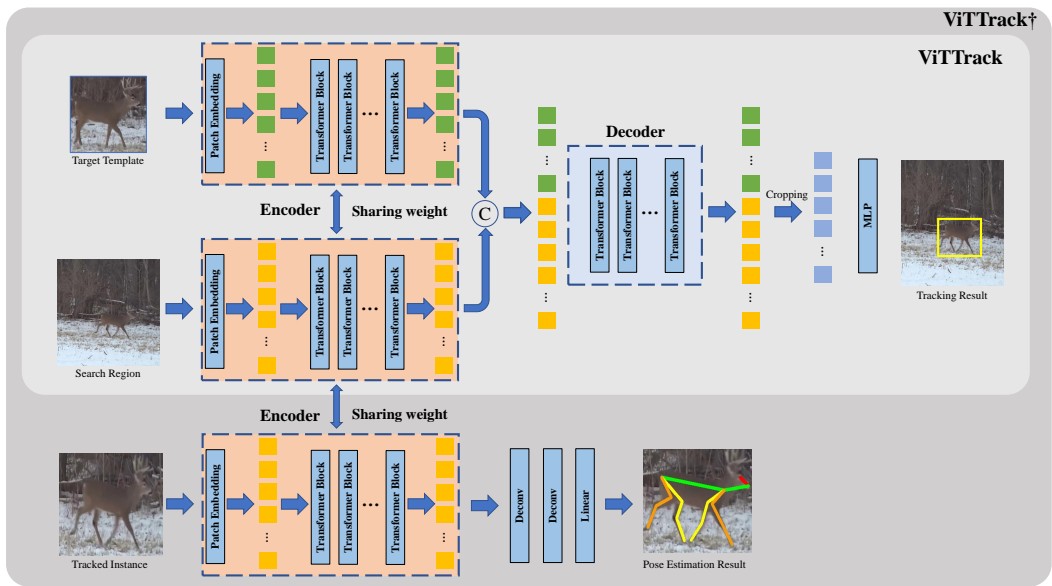

Figure 2: The structures of the proposed ViTTrack and ViTTrack†. We employ a plain vision transformer based encoder-decoder structure for object tracking in ViTTrack. To explore the potential of vision transformers for multi-task learning, we design a simple ViTTrack† model by using a shared backbone for pose estimation and tracking.

256*256 during both training and inference. The results are summarized in Table 2. It can be observed that the HRFormer [10] variants consume much more memory compared with other methods due to the quadratic memory consumption of transformer layers and the high-resolution feature map employed in their structures. The CNN-based framework SimpleBaseline [7] favors the inference speed and performs a little worse on the pose estimation accuracy than other methods. ViTPose obtains a better trade-off between the inference speed and accuracy, *i.e.*, it obtains 78.3 AP with 711 frame-per-second (FPS), while HRNet-w48 [6] obtains 77.4 AP with a lower 512 FPS.

## E   More implementation details

We train all the pose estimation models on 8 Nvidia A100 GPUs for 210 epochs, with the implementation details provided in the main text. We also provide a detailed description of the proposed ViTTrack framework in this section. As demonstrated in Figure 2, ViTTrack takes a siamese structure to extract features from the template and the search region with a weight-sharing encoder. The encoder is a plain vision transformer structure initialized with MAE pretraining on the ImageNet-1K training set. Then, the template and search region features are concatenated and fed into the decoder, which is also composed of several transformer layers with self-attention. To fully utilize the power of pretraining, we adopt exactly the same structure for the decoder from the MAE pretraining and initialize it accordingly. It is a reasonable design as the target of the decoder is modeling the similarity between input tokens no matter in the upstream masked image pretraining tasks and downstream visual object tracking task, *i.e.*, modeling the similarity between the template and the search regions. After that, the tokens corresponding to the search regions are selected and used to predict the location of the tracked instance. The ViTTrack model has been trained for 300 epochs with the GOT-10K [3] dataset, following the same setting in SwinTrack [4]. For the ViTTrack† framework, we first train the ViTPose model on the APT-36K dataset for 210 epochs with the animal pose estimation data. Then, the backbone weights of ViTPose are utilized to initialize the encoder in ViTTrack as they are the same ViT-B structure. After that, we train the ViTTrack's decoder with the GOT-10K dataset for 300 epochs, with the encoder part frozen. Thus, the pose estimator and tracker share the same encoder for feature extraction and separately use task-specific decoders for pose estimation and tracking. We denote the multi-task framework as ViTTrack†.

# F Datasheet

## F.1 Motivation

**1. For what purpose was the dataset created? Was there a specific task in mind? Was there a specific gap that needed to be filled? Please provide a description.**

**A1:** APT-36K is created to facilitate the development and evaluation of video-based animal pose estimation and tracking methods. Previous datasets either focused on animal tracking or single-frame-based animal pose estimation, and neither did both. Based on the proposed APT-36K dataset, we try to answer several challenging questions: 1) whether previous human or animal pose datasets benefit animal pose tracking tasks; 2) How well do animal pose estimation methods generalize across species in a balanced animal distribution setting; and 3) whether the animal pose estimation methods work well in animal pose tracking setting. The answer to the animal pose tracking task is impossible to explore with previous datasets as they do not consider the animal pose tracking scenarios. Besides, APT-36K is the first animal pose estimation dataset with balanced animal species distribution, with abundant species corresponding to long-tailed species in previous datasets [9]. Thus, we believe APT-36K can contribute to the development of animal pose tracking algorithms and single-frame-based animal pose estimation.

**2. Who created this dataset (e.g., which team, research group) and on behalf of which entity (e.g., company, institution, organization)?**

**A2:** APT-36K is created by the authors as well as some volunteer graduate students from Hangzhou Dianzi University, including Jiahao Jiang, Xuepu Zeng, Jiamian Xu, Jiacheng Zhang, Yuhu Xin, Jiajie Huang, Jingsheng Fang.

**3. Who funded the creation of the dataset? If there is an associated grant, please provide the name of the grantor and the grant name and number.**

**A3:** The project is funded by the National Natural Science Foundation of China under Grant 61873077.

## F.2 Composition

**1. What do the instances that comprise the dataset represent (e.g., documents, photos, people, countries)? Are there multiple types of instances (e.g., movies, users, and ratings; people and interactions between them; nodes and edges)? Please provide a description.**

**A1:** APT-36K consists of 2,400 video clips collected and filtered from 30 animal species with 15 frames for each video, resulting in 36,000 frames in total. The animal species are organized under taxonomic ranks. For each animal instance in the videos, we annotate its location, tracking identifier, and 17 keypoints (left eye, right eye, nose, neck, tail root, left shoulder, left elbow, left front paw, right shoulder, right elbow, right front paw, left hip, left knee, left hind paw, right hip, right knee, right hind paw). We also annotate the type of the background in the frames. The annotations are organized in the MS COCO [5] format.

**2. How many instances are there in total (of each type, if appropriate)?**

**A2:** The APT-36K dataset contains 36,000 images and 53,006 instances with high-quality position, tracking identifier, and keypoint annotations.

**3. Does the dataset contain all possible instances or is it a sample (not necessarily random) of instances from a larger set? If the dataset is a sample, then what is the larger set? Is the sample representative of the larger set (e.g., geographic coverage)? If so, please describe how this representativeness was validated/verified. If it is not representative of the larger set, please describe why not (e.g., to cover a more diverse range of instances, because instances were withheld or unavailable).**

**A3:** APT-36K collects and filters the videos from the YouTube website and covers 30 animal species selected from the real-world animal species. Due to the high diversity and huge amounts of animal species in the real world, it is almost impossible to include all animal species in a single dataset. We will continue to increase the diversity and volume of APT-36K in future work.

**4. What data does each instance consist of? "Raw" data (e.g., unprocessed text or images)or features? In either case, please provide a description.**

**A4:** Each instance consists of one animal with its RGB descriptions and annotations, including location (bounding box) annotation, family and species annotations, keypoints annotations, background type and tracking identifier annotations.

**5. Is there a label or target associated with each instance? If so, please provide a description.**

**A5:** Yes. Each instance is labeled with instance ID, image ID, animal species information (family and species), area of location box, keypoint information, number of keypoints, background category, the source of images and videos, and its unique tracking ID in the video, according to COCO labeling style.

**6. Is any information missing from individual instances? If so, please provide a description, explaining why this information is missing (e.g., because it was unavailable). This does not include intentionally removed information, but might include, e.g., redacted text.**

**A6:** Yes. We omit the possible low-quality annotations for some instances' keypoints, which may be heavily obscured, blurred, or small in scale. It is a common practice to improve the annotation quality as described in establishing the COCO human pose dataset. Following the labeling routine in MS COCO, we mark the confidence of these keypoints as zero in the annotation files.

**7. Are relationships between individual instances made explicit (e.g., users' movie ratings, social network links)? If so, please describe how these relationships are made explicit.**

**A7:** Yes. The annotations are labeled following the MS COCO labeling routine. Thus, the relationship between each instance, *e.g.*, whether two instances are located on the same image, can be queried using the COCO APIs.

**8. Are there recommended data splits (e.g., training, development/validation, testing)? If so, please provide a description of these splits, explaining the rationale behind them.**

**A8:** Yes. The dataset is split into three disjoint subsets for training, validation, and test, respectively, following the ratio of 7:1:2 per animal species. It is also noteworthy that we adopt a video-level partition to prevent the potential information leakage since the frames in the same video clip are similar to each other.

**9. Are there any errors, sources of noise, or redundancies in the dataset? If so, please provide a description.**

**A9:** Although we have double-checked the annotation information very carefully, there may be some inaccurate keypoint annotations, such as minor drift in the keypoint positions, due to the occlusion or blur caused by animals' moving.

**10. Is the dataset self-contained, or does it link to or otherwise rely on external resources (e.g., websites, tweets, other datasets)? If it links to or relies on external resources, a) are there guarantees that they will exist, and remain constant, over time; b) are there official archival versions of the complete dataset (i.e., including the external resources as they existed at the time the dataset was created); c) are there any restrictions (e.g., licenses, fees) associated with any of the external resources that might apply to a future user? Please provide descriptions of all external resources and any restrictions associated with them, as well as links or other access points, as appropriate.**

**A10:** Yes. APT-36K does not use content from other existing datasets. The video clips used in APT-36K are from publicly available videos on the Youtube dataset. We acknowledge the video creators' efforts and attach the links to the source videos in the APT-36K dataset.

**11. Does the dataset contain data that might be considered confidential (e.g., data that is protected by legal privilege or by doctorpatient confidentiality, data that includes the content of individuals non-public communications)? If so, please provide a description.**

**A11:** No.

**12. Does the dataset contain data that, if viewed directly, might be offensive, insulting, threatening, or might otherwise cause anxiety? If so, please describe why.**

**A12:** No.

## F.3 Collection Process

**1. How was the data associated with each instance acquired? Was the data directly observable (e.g., raw text, movie ratings), reported by subjects (e.g., survey responses), or indirectly inferred/derived from other data (e.g., part-of-speech tags, model-based guesses for age or language)? If data was reported by subjects or indirectly inferred/derived from other data, was the data validated/verified? If so, please describe how.**

**A1:** The keypoint information is observable for each animal. Well-trained annotators are recruited to label the locations of the animal keypoints directly with annotation tools like labelme.

**2. What mechanisms or procedures were used to collect the data (e.g., hardware apparatus or sensor, manual human curation, software program, software API)? How were these mechanisms or procedures validated?**

**A2:** The video clips in APT-36K are all selected and filtered from publicly available videos on Youtube. We manually select the representative clips from these videos, where each clip contains observable animal movement.

**3. If the dataset is a sample from a larger set, what was the sampling strategy (e.g., deterministic, probabilistic with specific sampling probabilities)?**

**A3:** No.

**4. Who was involved in the data collection process (e.g., students, crowdworkers, contractors) and how were they compensated (e.g., how much were crowdworkers paid)?**

**A4:** The authors of the paper.

**5. Over what timeframe was the data collected? Does this timeframe match the creation timeframe of the data associated with the instances (e.g., recent crawl of old news articles)? If not, please describe the timeframe in which the data associated with the instances was created.**

**A5**: It took about 15 days to collect the data and about 3 months to complete the data cleaning, organization, and annotation process.

## F.4 Preprocessing/cleaning/labeling

**1. Was any preprocessing/cleaning/labeling of the data done (e.g., discretization or bucketing, tokenization, part-of-speech tagging, SIFT feature extraction, removal of instances, processing of missing values)? If so, please provide a description. If not, you may skip the remainder of the questions in this section.**

**A1:** We manually collect and clean the data from the YouTube website to ensure there are high-quality images in the APT-36K dataset with 1920*1080 resolution. Besides, we have each annotator manually check videos containing several animal instances to remove the duplicate videos in the dataset.

**2. Was the "raw" data saved in addition to the preprocessed/cleaned/labeled data (e.g., to support unanticipated future uses)? If so, please provide a link or other access point to the "raw" data.**

**A2:** No. We provide the links to the source videos in the annotation files.

**3. Is the software used to preprocess/clean/label the instances available? If so, please provide a link or other access point.**

**A3:** We use the open source labeling tool labelme.

## F.5 Uses

**1. Has the dataset been used for any tasks already? If so, please provide a description.**

**A1:** No.

**2. Is there a repository that links to any or all papers or systems that use the dataset? If so, please provide a link or other access point.**

**A2:** N/A.

**3. What (other) tasks could the dataset be used for?**

**A3:** APT-36K can be used for both animal pose estimation and animal pose tracking studies. In addition, it can be used for topics such as animal species classification as well as animal behavior understanding and behavior prediction (with further corresponding annotations).

**4. Is there anything about the composition of the dataset or the way it was collected and preprocessed/cleaned/labeled that might impact future uses? For example, is there anything that a future user might need to know to avoid uses that could result in unfair treatment of individuals or groups (e.g., stereotyping, quality of service issues) or other undesirable harms (e.g., financial harms, legal risks) If so, please provide a description. Is there anything a future user could do to mitigate these undesirable harms?**

**A4:** No.

**5. Are there tasks for which the dataset should not be used? If so, please provide a description.**

**A5:** No.

## F.6 Distribution

**1. Will the dataset be distributed to third parties outside of the entity (e.g., company, institution, organization) on behalf of which the dataset was created? If so, please provide a description.**

**A1:** Yes. The dataset will be made publicly available to the research community.

**2. How will the dataset will be distributed (e.g., tarball on website, API, GitHub)? Does the dataset have a digital object identifier (DOI)?**

**A2:** It will be publicly available on the project website at GitHub.

**3. When will the dataset be distributed?**

**A3:** The dataset will be distributed once the paper is accepted after peer-review.

**4. Will the dataset be distributed under a copyright or other intellectual property (IP) license, and/or under applicable terms of use (ToU)? If so, please describe this license and/or ToU, and provide a link or other access point to, or otherwise reproduce, any relevant licensing terms or ToU, as well as any fees associated with these restrictions.**

**A4:** It will be distributed under the CC-BY-4.0 licence.

**5. Have any third parties imposed IP-based or other restrictions on the data associated with the instances? If so, please describe these restrictions, and provide a link or other access point to, or otherwise reproduce, any relevant licensing terms, as well as any fees associated with these restrictions.**

**A5:** No.

**6. Do any export controls or other regulatory restrictions apply to the dataset or to individual instances? If so, please describe these restrictions, and provide a link or other access point to, or otherwise reproduce, any supporting documentation.**

**A6:** No.

## F.7 Maintenance

**1. Who will be supporting/hosting/maintaining the dataset?**

**A1:** The authors.

**2. How can the owner/curator/manager of the dataset be contacted (e.g., email address)?**

**A2:** They can be contacted via email available on the project website.

**3. Is there an erratum? If so, please provide a link or other access point.**

**A3:** No.

**4. Will the dataset be updated (e.g., to correct labeling errors, add new instances, delete instances)? If so, please describe how often, by whom, and how updates will be communicated to users (e.g., mailing list, GitHub)?**

**A4:** No. Since we conduct three-round annotation check, we believe the annotation errors are very seldom in our APT-36K dataset. The occasional mistakes such as annotation drift can be treated as noise in the dataset.

**5. Will older versions of the dataset continue to be supported/hosted/maintained? If so, please describe how. If not, please describe how its obsolescence will be communicated to users.**

**A5:** N/A.

**6. If others want to extend/augment/build on/contribute to the dataset, is there a mechanism for them to do so? If so, please provide a description. Will these contributions be validated/verified? If so, please describe how. If not, why not? Is there a process for communicating/distributing these contributions to other users? If so, please provide a description.**

**A6:** N/A.