# OpenReview forum: "APT-36K: A Large-scale Benchmark for Animal Pose Estimation and Tracking"
_NeurIPS.cc/2022/Track/Datasets_and_Benchmarks — NeurIPS 2022 Datasets and Benchmarks _

### Official Review · Reviewer_pfis · 2022-07-25
**Technical Review**

**Rating:** 7
**Confidence:** 4
**Correctness:** They seem correct to me.
**Clarity:** The paper is very well written.

**Strengths:**

The paper is well written and easy to follow.
The authors set up three different tracks on this dataset, including SF, IS, and APK tracks. Experiments seem technically sound.


**Weaknesses:**

My only concern is the novelty of this dataset on animal pose estimation. AFAIK, the pose estimation in the animal kingdom dataset [26] consists of 33K frames, corresponding to 850 animal species. The animal kingdom dataset has more animal species with the same amount of frames, so the reviewer thinks the animal kingdom dataset is more challenging than the proposed dataset on animal pose estimation. Since the proposed dataset contains one more animal pose tracking task, I prefer to put the rating at 'Marginally above acceptance threshold'.



**Additional Feedback:**

It is better to provide detailed information about the animal species and animal families.

**Documentation:**

As the GitHub page is invalid, the reviewer thinks this dataset seems to be not well documented.

**Ethics:**

N.A.

**Relation To Prior Work:**

Its relation to prior works is clearly discussed.

**Summary And Contributions:**

This paper introduces a new dataset (APT-36K) for animal pose estimation and tracking. APT-36K consists of 2400 video clips with 30 animal species. The authors set up three tracks (SF, IS, APK) based on their collected dataset.

---

### Official Review · Reviewer_V7NX · 2022-07-25
**A useful dataset that can be valuable resource for video-based animal pose estimation.**

**Rating:** 6
**Confidence:** 5
**Correctness:** I do not have any concerns about the …
**Clarity:** The paper is well written.

**Strengths:**

-Paper is well written, dataset statistics and data collection details are clearly mentioned.

-Multiple SOTA pose estimation and tracking approaches are evaluated. Authors report performance under various weight initialisation and multiple backbones, providing useful benchmarks for further work.

-Luck of large data collections remains a limiting factor for animal pose estimation research, especially given the existence of arbitrary many animal species and the large appearance and shape variation for different animals. Large-scale data collections can be valuable for enabling further research in the area.

**Weaknesses:**

-On the SF track, authors report model performance overall captured animal species. The reader could potentially benefit from a more granular analysis, showing performance per individual species. This can lead to a better understanding of how SOTA models can capture different animal categories. Are some species more challenging than others?

-In the IS track, authors evaluate the generalisation abilities of an HRnet32 on unseen animal species. The outcome of this analysis is rather intuitive. A supervised pose estimation model would have a significant performance decrease on an unseen object category. Is inter-species generalisation expected by supervised models? For example, the semantics of keypoints might change for different animals (as stated in lines 166-168). Even if a model is rather consistent in capturing an unseen category, it is possible that it can follow a slightly different keypoint configuration than the one manually defined in the ground-truth (i.e consistently capturing the same semantic point in slightly different object location leading to large error values). I am concerned about the significance of this analysis or its suitability for the comparison of different models. A more appropriate path for evaluating the generalisation of different models could be through a transfer or few shot learning pipeline. Related experimentation was included in [38] for the APT-10K dataset.

-The claim “well-trained annotators” might be a little vague and some more details regarding the cross-checking performed could be informative.

-As also mentioned by the Authors, this dataset is still relatively small compared to human pose estimation benchmarks. Particularly given the large number of animals captured, leading to a few thousand images per animal species.

**Additional Feedback:**

Overall I feel this is a useful dataset and can be valuable for further research on video-based animal pose estimation.

**Documentation:**

Dataset is well documented.

**Ethics:**

I do see any ethical concerns about this paper.

**Relation To Prior Work:**

The contribution of this work compared to previous art is clearly discussed

**Summary And Contributions:**

In this work, APT-36K is introduced, a new dataset for animal pose estimation and tracking. APT-36K consists of 36K frames, sampled from youtube videos and manually annotated for 53k animal instances. In total, 30 animal species are captured with 80 video clips per species, making this a balanced set. A significant annotation effort is carried out, as APT-36K includes animal keypoints, bounding boxes, instance level ids and background class labels. Dataset statistics are thoroughly presented and baseline performance for SOTA pose estimation models is reported. Given the lack of video-based datasets for animal pose estimation, APT-36K can be valuable for further research.

---

### Official Review · Reviewer_1Y5Z · 2022-07-27
**Nice dataset, but concerns about licensing**

**Rating:** 6
**Confidence:** 3
**Correctness:** I believe the claims and dataset cons…
**Clarity:** Yes, the clarity of the manuscript is…

**Strengths:**

The paper is clear, the dataset and ethics are well documented, and the dataset is of good value to the community. M

**Weaknesses:**

My main reservations are on the novelty of the dataset and the licensing.


Data licensing. The data is described as originating from YouTube . According to the Youtube terms of service (https://www.youtube.com/static?template=terms)
>“You are not allowed to:
1.	access, reproduce, download, distribute, transmit, broadcast, display, sell, license, alter, modify or otherwise use any part of the Service or any Content except: (a) as expressly authorized by the Service; or (b) with prior written permission from YouTube and, if applicable, the respective rights holders;
“

Moreover, it is unclear if the particular videos are licensed under the youtube standard license or a CC-BY license, which would affect whether or not they could be distributed and modified, even for noncommercial use. I am not a legal expert, but my understanding is that this dataset may be in violation of (1) The YouTube service agreement (pertinent, since Alphabet is typically a Neurips sponsor) and (2) the copyright holders of these video, unless YouTube has specifically granted permission for their use.


Novelty: The AP-10K dataset was at Neurips-DB 2021, and this dataset is fairly similar. There are key differences – the data here is better balanced around a few categories, and these are videos, but many of the tasks used are pose estimation in animals.

There are occlusions and tracking errors in some clips – is there a ground truth upper bound on the possible mAP in this dataset?


**Additional Feedback:**

Minor


Can you provide a statement on how animal species were chosen and video clips? Several of the video clips appear to have animals wearing clothes, which seems outside the scope.

Table 1 – What does N/A denote? For instance it seems the Horses dataset should have ‘1’ family.
Table 3 – should the title say AP-10K or APT-36k?
“According to Darwin’s theory of evolution, animals belonging to the same taxonomic rank share some similarities in behavior patterns, anatomical keypoint distribution, and appearance.” This isn’t really accurate, as natural selection is distinct from linnean taxonomy, and even in e.g. Dogs you can have large behavioral and phenotypic variation.


**Documentation:**

Yes.

**Ethics:**

Yes, I have concerns about the dataset licensing, although I am not an expert here.

**Relation To Prior Work:**

Firsts: The authors frequently claim to be the ‘first’ for various things, e.g. “the first large-scale dataset with high-quality animal pose annotations from consecutive frames for animal pose estimation and tracking.” In general this sort of credit claiming is discouraged, and while I do think this paper is potentially the first to propose a tracking baseline, but there are other animal datasets with continuous frames, e.g. the Multi-agent behavior challenge (CalMS), the Open Monkey Studio datasets, and the PAIRS R-24 dataset, I would qualify these claims.


**Summary And Contributions:**

This paper describes the APT-36k dataset, consists of 2,400 video sequences, each on 16 frames of annotated video, with applications to video tracking and pose estimation. The videos are balanced across 15 families of animals (30 species), allowing for estimation of cross-family generalization. The authors’ give benchmarks for the dataset in three categories – single frame pose estimation, cross-species generalization, and pose tracking, where an object tracking approach is needed to identify the bounding box in each frame. The paper is clear and well-written, and the datasheet and methods are well described.

I am enthusiastic about this manuscript as a resource for the community. However, I have several reservations. First and foremost, as described below, I am not sure it is permissible to download youtube videos that make up the dataset. Because of this I am voting not to accept the manuscript, but if this issue is reconciled I would consider accepting. Second, the dataset is more of an incremental advance over last year’s AP-10K. This does not preclude its acceptance, but does reduce its novelty for the community. Of the three tasks presented, only one requires the continuous video data uniquely included here.

---

### Official Review · Reviewer_dKLj · 2022-07-27

**Rating:** 7
**Confidence:** 4
**Correctness:** Yes
**Clarity:** Yes

**Strengths:**

+ Large-scale annotations.
+ Detailed statistics of the APT-36K dataset are provided. The animal motions are diverse.
+ Comprehensive experiments are conducted to benchmark 3 different tracks.

**Weaknesses:**

- The authors should provide the details of metric AP. Since AP is calculated using the metric object keypoint similarity (OKS), which needs to define the per-keypont constant `k_i` for each keypoint `i`. `k_i` controls the falloff and varies among different keypoints.
- For each video clip, the authors sample 15 frames for annotations and the sample rate depends on the animal motions. I suggest the authors provide statistics about the video length. This is important for pose tracking since a longer video is more difficult to track.
- The authors should use more metrics for pose tracking. There are many standard metrics for pose tracking, such as MOTA, MOTP, ID Switch.

**Additional Feedback:**

- In Table 2, I notice that existing methods obtain quite good performance on single frame pose estimation. Does it suggest that APT-36K is not difficult enough for pose estimation?

**Documentation:**

Yes

**Ethics:**

Yes

**Relation To Prior Work:**

Yes

**Summary And Contributions:**

This work presents a new benchmark, APT-36k, for animal pose estimation and tracking. APT-36K consists of 2400 video clips and 36000 frames for 30 animal species. It is the first benchmark for both animal pose estimation and pose tracking. Based on APT-36k, the authors benchmark three different tasks: 1. single-frame animal pose estimation; 2. inter-species domain generalization; 3. animal pose estimation and tracking.

---

### Official Review · Reviewer_a5RF · 2022-07-28
**APT-36K: A Large-scale Benchmark for Animal Pose Estimation and Tracking**

**Rating:** 6
**Confidence:** 4
**Clarity:** Yes. Overall, the paper is well organ…

**Strengths:**

1. The proposed APT-36K is the first dataset for animal pose estimation and tracking. The scale of the APK-36K is relatively large compared with existing animal pose estimation datasets. The dataset is carefully collected to cover different animal species from different scenes.

2. Three tasks are set up to evaluate several representative models. The results and analysis provide some insights which are useful for researchers in this field. For the single frame pose estimation task, several representative CNN and transformer-based models are compared and also the impact of pre-training is investigated. The experiments for the inter-species animal pose generalization task show that to achieve a good generalization performance for a specific species, it is usually necessary to collect some training data of species which belong to the same family.

**Weaknesses:**

1. For the inter-species animal pose generalization task, only 6 representative animal families are used. What would the results and conclusion be when all the species are included?

2. For the animal pose tracking task, only single object tracking methods are evaluated. Some multiple object tracking methods should also be included for more thorough comparison. Typically, some human pose estimation and tracking methods (e.g. SimpleBaseline [34]) are needed to be evaluated.


**Additional Feedback:**

See weaknesses.

**Correctness:**

The dataset is constructed in a sound way. The evaluation methods and experiment design are appropriate.

**Documentation:**

It seems that the github links do not work.

**Relation To Prior Work:**

Yes

**Summary And Contributions:**

The paper presents a dataset, APT-36K, for animal pose estimation and tracking. APT-36K consists of 2,400 video clips collected and filtered from 30 animal species with 15 frames for each video, resulting in 36,000 frames in total. Based on APT-36K, three tasks are set up: (1) single frame pose estimation, (2) inter-species domain generation test and (3) animal pose estimation and tracking. Several representative models are evaluated on the three tasks.

Main contributions: (1) a large-scale benchmark dataset APT-36K for animal pose estimation and tracking; (2) the setup of three tasks with several representative models benchmarked.

---

### Meta-Review · Program_Chairs · 2022-09-16

**Recommendation:** Accept
**Confidence:** 4

**Metareview:**

After careful review, I believe that this paper is a useful contribution to the study of animal pose estimation and tracking. The paper received positive reviews from all the reviewers and the authors have successfully addressed the concerns regarding the lack of sufficient novelty and insufficient benchmarking. The authors also added additional low shot experiments to demonstrate the usefulness of the dataset. Based on this, I think the paper meets the bar for the track and should be accepted.

---

### Decision · Program_Chairs · 2022-09-16

Accept